# Early Nutritional Education in the Prevention of Childhood Obesity

**DOI:** 10.3390/ijerph18126569

**Published:** 2021-06-18

**Authors:** Mario Gato-Moreno, María F. Martos-Lirio, Isabel Leiva-Gea, M. Rosa Bernal-López, Fernando Vegas-Toro, María C. Fernández-Tenreiro, Juan P. López-Siguero

**Affiliations:** 1Department of Pediatric Endocrinology, Regional University Hospital of Málaga, 29011 Málaga, Spain; mariogatomoreno@gmail.com (M.G.-M.); mariaml_huelma@hotmail.com (M.F.M.-L.); fevetor@gmail.com (F.V.-T.); mencyta@hotmail.com (M.C.F.-T.); lopez.siguero@gmail.com (J.P.L.-S.); 2Instituto de Investigación Biomédica de Málaga (IBIMA), 29010 Málaga, Spain; robelopajiju@yahoo.es; 3Department of Pharmacology and Pediatrics, Faculty of Medicine, University of Málaga, 29016 Málaga, Spain; 4Department of Internal Medicine, Regional University Hospital of Málaga, 29009 Málaga, Spain; 5CIBER Fisiopatologia de la Obesidad y la Nutricion (Ciber Obn), Carlos III Health Institute, 28029 Madrid, Spain; 6Ministry of Education and Sports, 29002 Málaga, Spain

**Keywords:** body mass index, overweight, childhood obesity, preschool, early intervention, educational, healthy diet, prevention, pediatric obesity

## Abstract

Early childhood is a critical period for obesity prevention. This randomized controlled study evaluated the effectiveness of an educational intervention preventing obesity in preschool-age children. A nutritional education intervention, with a follow-up session one year later, was conducted with parents of children aged 3 to 4 years of public schools in the province of Málaga. The main outcome variable was the body mass index z-score (zBMI). The prevalence of overweight or obesity was the secondary outcome variable. The sample comprised 261 students (control group = 139). Initial BMI, weight, height-for-age and prevalence of overweight and obesity were similar for both groups. After the first year of the intervention, the zBMI of the intervention group decreased significantly from 0.23 to 0.10 (*p* = 0.002), and the subgroup of patients with baseline zBMI above the median decreased from 1 to 0.72 (*p* = 0.001), and in the second year from 1.01 to 0.73 (*p* = 0.002). The joint prevalence of overweight and obesity increased in the control group (12.2% to 20.1%; *p* = 0.027), while in the intervention group, there were no significant changes. This preschool educational intervention with parents improved their children’s BMI, especially those with a higher BMI for their age, and favored the prevention of overweight or obesity.

## 1. Introduction

In recent decades, the increase in the prevalence of childhood obesity has become a worldwide public health problem [1,2]. Recent studies in the United States analyzed the prevalence of obesity in children and found that it increases with age such that at 14 years of age, 20.8% of children are obese and 17% are overweight [3]. In Spain, the Aladino study (2019), which analyzed the national prevalence of obesity and overweight in schoolchildren aged 6 to 9 years, showed worrying figures: 40.6% of the schoolchildren analyzed were overweight. Of this percentage, 23.3% were overweight and 17.3% obese [4]. Similarly, an Andalusian study that assessed growth and prevalence of overweight and obesity in children [5] found the highest prevalence of overweight at 12 years old (26.8%) and of obesity at 8 years old (14%).

Different risk factors for obesity have been described, including male sex [6], obese parents [7], low socioeconomic status, high birth weight for gestational age [8], artificial formula feeding [9], rapid weight gain in the first months of life, excessive protein intake [10], early adiposity rebound [11], unhealthy dietary behaviors and sedentary lifestyle [12,13]. Childhood obesity is associated with multiple complications such as lipid metabolism disorders, hypertension, hyperinsulinism [14], hepatic steatosis, obstructive sleep apnea syndrome [15], chronic inflammation [2] and psychological problems [16].

Certain periods of childhood are crucial for obesity prevention, as they are associated with notable changes in adiposity. These periods include the first 2 years of life, the period of adiposity rebound (between 5 and 7 years of age) and puberty [17]. Early childhood is a critical period for the development of obesity [18]. Indeed, the increase in body mass index (BMI) begins at around 4 or 5 years of age, and it has been shown that children who are overweight at 5 years of age are 4 times more likely to become obese later in life [3]. Once excess weight is established, it is difficult to reverse and the probability of becoming obese in adulthood increases, which makes early intervention necessary [19]. The prevention of childhood obesity is an international health priority and should be initiated in the prenatal period, maintaining appropriate weight gain during pregnancy. It is also advisable to introduce proper breastfeeding and complementary feeding, promote active play, avoid a sedentary lifestyle, ensure sufficient sleep [20] and adhere to the Mediterranean diet, which has been shown to reduce the risk of overweight and obesity [21].

The main objective of intervention programs to prevent childhood obesity is to improve nutrition and physical activity in the youngest children, since the rate of failure and relapse in established obesity is high [22]. These programs should have the following characteristics: broad scope, cost-effective policy, social perspective, and a multi-sectoral and multi-level approach [20,23].

The school is an ideal place to carry out lifestyle interventions, since schoolchildren spend a large number of hours in this setting [24]. Additionally, it allows access and a close contact with a population that is already gathered and will likely be available over time. The most effective interventions are those that combine diet with physical exercise in children aged 0 to 5 years [25]. School-based interventions have been found to be effective in reducing the BMI [26] of children, and this intervention, when implemented in the home, can even improve the BMI of their parents [27]. The aim of this study was to determine the impact of a school-based educational intervention on eating behavior and physical activity aimed at the parents of children aged 3 to 4 years and on the evolution of BMI and the prevalence of overweight or obesity in their children in the 2 years following the intervention. Our hypothesis is that the preschool-aged children whose parents receive an educational intervention can improve their BMI.

## 2. Materials and Methods

This was a school-based parent-only intervention study involving an education program for parents of children at the beginning of their schooling and was designed as a randomized clinical trial with a control group. We included parents of students aged 3 to 4 years in public schools in the province of Málaga, Spain, who agreed to participate in the study (n = 261). None of the students had a serious chronic disease with a high risk of malnutrition that was a reason for exclusion. A single-stage cluster sampling (centers) stratified by counties was undertaken so that all areas of the province of Málaga were represented. Then, a stratified randomization of these centers considering rural and urban areas of Málaga was performed to establish the intervention group (IG, n = 122) and the control group (CG, n = 139). The schools formed units assigned to the same group, that is, each complete center was either intervention or control. Baseline homogeneity between groups was confirmed. The duration of the study was 2 years.

The intervention was carried out by a dietetic technician in collaboration with school physicians (Educational Guidance Teams). In the IG, the first intervention was delivered at the beginning of the first year through a set of six 2 h group training sessions, given every two weeks; and a follow-up intervention at the beginning of the second year through a 3 h session. In these six sessions, group and interactive activities were used to address the following topics: introduction to nutrition, healthy eating habits, keys to improving nutrition and eating habits, designing healthy and attractive menus, physical activity and food labeling. In the annual follow-up session, the topics of nutrients and their importance, healthy menus and physical activity were revisited. Sessions were aimed to parents, thus children were not present. These interventions are described in Appendix A.

### 2.1. Variables

Auxological assessment was performed at the beginning of the first year of schooling (time 0) and at the end of the first and second years (times 1 and 2). Weight was determined using a Taurus Oslo model scale with a maximum weight capacity of 150 kg and accurate to 0.1 kg, recording the mean of two consecutive measurements. For weight measurement, the subjects were barefoot and wearing light clothing. Standing height was measured with a Seca 213 portable stadiometer, accurate to 0.1 cm. The mean of two consecutive measurements was recorded. The measurements were in kg and cm, respectively, to one decimal place. Quantitative variables were expressed as means ± standard deviation (SD) and dichotomous variables as percentages.

The response variable was the variation in BMI z-score (zBMI) using the reference values for sex and age from Spanish Growth Studies 2010 [28]. The zBMI was used to measure changes in the children since, unlike the absolute value of other variables such as weight or BMI, this variable reflects more accurately changes in a child’s body composition over time and is adjusted according to sex, independently of the increase in weight due to growth. The secondary variable was the prevalence of overweight or obesity. Overweight was considered to be a BMI between the 85th and 95th percentiles and obesity equal to or above the 95th percentile [29].

### 2.2. Statistical Analysis

The data were analyzed with R 4.0.2 software (R Core Team 2020). The fit of the variables to the normal distribution was tested with the Shapiro–Wilk test. The only variable that followed a normal distribution and used the student’s t-test was zHeight. The Mann–Whitney U test was used in the rest of the quantitative variables of Table 1. The Wilcoxon signed-rank test was used for quantitative variables of Table 2 and Table 3 in related samples that did not follow a normal distribution. To compare variables expressed as percentages, the χ² test was used in independent samples (Table 1) and McNemar’s test in related samples (Table 4). A *p* value of less than 0.05 was considered statistically significant. Discrete variables were expressed as percentages and continuous variables as mean ± standard deviation or median (interquartile range), depending on whether or not they conformed to the normal distribution. An “intention to treat” analysis strategy was utilized. The statistical analysis and results were validated by an independent professional statistical team (Biostatech).

## 3. Results

A total of 261 students were included with a mean age of 44.9 ± 4.63 months and 46.4% were female. The CG comprised 139 students and the IG 122. The flow diagram of the participants is shown in Figure 1. The losses were associated with a change of school or failure to attend class on the day of the auxological assessment.

CG individuals were significantly younger (1.93 ± 0.56 months). No significant differences were found at baseline between the groups for the rest of the variables studied. Both groups had similar BMI, weight and height-for-age, with no differences by sex. Both groups also had a similar proportion of students with zBMI over the median (value = 0), and of overweight and/or obese individuals (see Table 1).

The changes observed in the zBMI in the CG and IG during the two years of this study are provided in Table 2. In the CG, the evolution of zBMI showed no significant changes in either year of the study. However, in the IG, a significant decrease in zBMI was observed both after the first year of the intervention (baseline 0.23 ± 1.18, final 0.10 ± 0.99; *p* = 0.002), and after comparing the baseline zBMI with that of the second year (baseline 0.24 ± 1.21; final 0.14 ± 1.05; *p* = 0.021). The variation observed in the number of baseline data from the first and second year is due to the loss of following-up some patients in the second year.

When analyzing the evolution of the zBMI stratified into subgroups according to whether the initial zBMI was lower or higher than the median of the total sample (with a value of 0), it was observed that in the subgroup of patients with zBMI lower than or equal to the median, the zBMI showed a similar evolution in both the CG and the IG, with no significant changes in any of them during the 2 years of the study. However, in the subgroup of patients with zBMI above the median, a different evolution was observed between the zBMI of the CG and the IG (see Table 3). While in the CG there were no significant differences in the evolution of the zBMI, in the IG there was a significant decrease of 0.28 points in the zBMI, both in the first year (baseline 1.00 ± 1.30, final 0.72 ± 1.05; *p* = 0.001) and in the second year (baseline 1.01 ± 1.32, final 0.73 ± 1.15; *p* = 0.002).

Regarding the evolution of the combined prevalence of overweight or obesity, no significant differences were found during the first year, in either the CG or the IG (see Table 4). However, when comparing the baseline prevalence with the prevalence after 2 years, a statistically significant increase of 7.6% in the prevalence of overweight and obesity was found in the CG (initial 12.2%, final 20.1%; *p* = 0.027), while in the IG, there were no significant changes.

## 4. Discussion

The main objective of this research was to evaluate the effectiveness of a training program given to parents of children 3 to 4 years of age on the evolution of BMI. In this study, the main effect observed was a decrease in zBMI in the IG, which did not occur in the CG. This effect occurred mainly in the first year after the intervention, and in children with a baseline zBMI above the median of the total sample.

This decrease in the zBMI in the IG one year after the intervention showed a clear benefit of the intervention with respect to the CG. One of the main advantages of this school-based intervention was the availability of a localized sample in schools, which facilitated the achievement of a cost-effective prevention intervention with a broad scope [24], including 70% of the children in the province aged 3 years. Other interventions carried out in Spanish schools, such as that performed by Pérez-Solís et al. [30] have also shown positive effects in reducing the zBMI (reduction between 1.14 and 1.02 in the IG) in children aged 6 years and older. However, the older age of the intervention subjects in the study by Pérez-Solís et al. allowed their intervention to be carried out directly with the children through workshops, talks and written material in the school, instead of only with the parents, as in our study.

Other interventions performed with parents of children of similar ages to those in our study have also found positive results, with a decrease in zBMI one year after receiving nutritional education [31]. The intervention by Slusser et al. showed greater effectiveness than ours (IG −0.20 versus CG +0.04), which could be associated with the longer duration of the intervention. The magnitude of the benefit is directly proportional to the time dedicated to parent education. Concerning this approach, a recent systematic review indicates that the most effective interventions are those that last more than 6 months with sessions at least every 2 weeks [32].

Despite the marked benefit of the intervention shown after the first year, the protective effect was diminished at 2 years. This suggests that the follow-up session at 1 year (3 h) may be insufficient to maintain adherence and may not have the same effect as the initial intervention. However, it is important to consider that this follow-up session was attended by only 22% of the IG parents compared to 100% who attended the initial intervention.

The subgroup analysis nonetheless showed that the intervention particularly benefited children with a zBMI higher than the median of the total sample, demonstrating a scope that not only prevented an increase in zBMI, but also helped to reduce these values. In addition, it was shown to significantly prevent an increase in the prevalence of overweight or obesity in the IG compared to the CG 2 years after the initial intervention. This protective effect for the increase in prevalence was particularly evident in the first year. From the first to the second year, a similar but not statistically significant increase in prevalence was observed in both study groups, most likely due to the loss of effect of the initial intervention and the poor adherence to the follow-up intervention in the IG. In this aspect, our study obtained better results than Pérez-Solís, who, after 2 years of intervention found no change in the prevalence of overweight or obesity between the CG and the IG.

The magnitude of the decrease in zBMI achieved with our intervention in children with higher zBMI was particularly significant when we consider that early childhood (3–6 years) is a key period for establishing habits and where overweight and obesity begin to appear. Multiple studies show that weight at 5 years of age is a good predictor of weight at 9 years of age [33,34], and could be associated with the development of future metabolic complications such as diabetes [35]. For this reason, the preschool stage may be the best time to implement strategies for the prevention of all conditions associated with overweight and obesity. It is also important to raise awareness among parents that overweight and obesity are increasing trends that generally do not disappear as children grow older, and therefore should be addressed as soon as possible [13,36].

In addition to the clear benefit demonstrated for the children, the nutritional education received by the parents may also result in an improvement in the eating habits of the other members of the family [37]. Although our study did not assess the effect that may have occurred in families, international studies of interventions at the family level to prevent childhood obesity have found that parents experience an improvement in their body composition as a secondary effect of the intervention [27].

The novel aspect of this study is that it is one of the first to evaluate the effectiveness of a nutritional education intervention in the population of preschool-aged children in southern Europe.

The main limitations of this study were the loss to follow-up of the participants, and the decreased adherence to the intervention in the annual follow-up phase. Ideally, the duration and frequency of the intervention sessions should be increased to improve their effectiveness. However, lack of parental adherence to the intervention could detract from the effectiveness of the intervention. Furthermore, no method was established to verify that the intervention covered all the planned aspects, such as the recording of the sessions carried out. Moreover, the baseline homogeneity of physical activity and eating habits between groups was not analyzed directly. Although the differences in physical activity levels generally are not very important at this age, we did analyze other baseline outcome variables that could be indirectly related to these behavioral parameters (zBMI, percentage zBMI > 0, percentage of overweight and/or obese students).

For future interventions, we recommend emphasizing the use of content adapted to foods available by season, area, customs and family economy, in order to promote better adherence and avoid the stigma that healthy eating is perceived as more expensive [30]. The emotional aspect derived from the relationship that individuals have with food should also be taken into account and addressed in the context of an efficient program focused on preserving health, not on weight loss as the ultimate goal. We also found that it would have been interesting to analyze the demographic characteristics of the parents to identify possible barriers that could hinder the intervention, in this way, strategies to overcome these barriers could be implemented.

## 5. Conclusions

This parent-only intervention program administered in the school setting at the initiation of early childhood education achieved a significant improvement in the BMI of the children whose parents received the educational intervention, especially in the children who started with a high BMI for their age. This type of educational intervention with such a young target population appears to be an effective and promising strategy to prevent excess weight and its long-term health consequences. Education on nutrition and physical activity should therefore be recommended as part of school activities from the age of 3 years with the participation of parents. Nevertheless, further studies in this field with larger sample sizes and longer follow-up times are needed to confirm and better delimit the beneficial effects demonstrated in our study, both in the short and long term in the life of the child.

## Figures and Tables

**Figure 1 ijerph-18-06569-f001:**
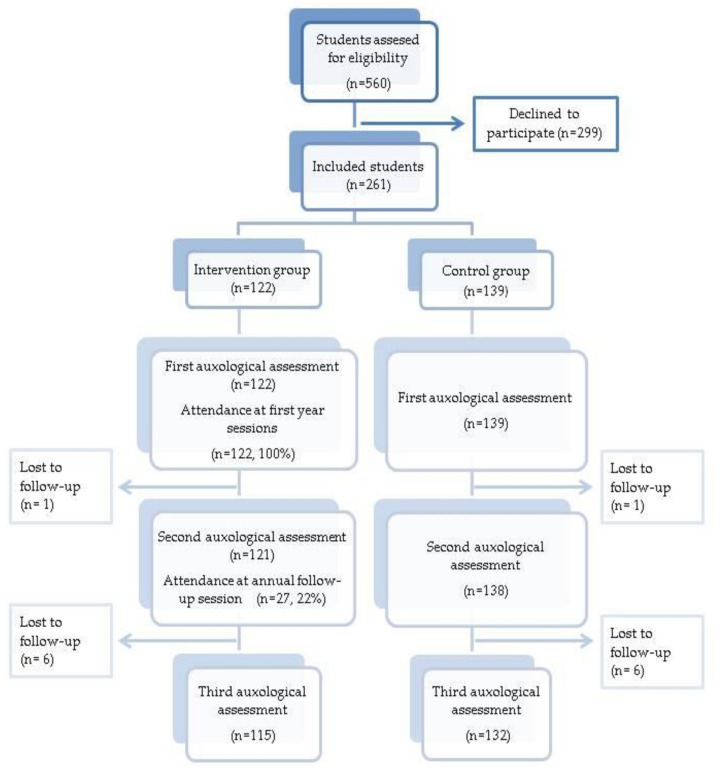
Study flow diagram.

**Table 1 ijerph-18-06569-t001:** Characteristics of the participants at the beginning of the study.

Variable	Control	Intervention	*p*
n	139	122	
Female, %	43.90	49.20	0.392
Age (months)	43.81 ± 3.93	45.70 ± 5.41	0.002 *
zBMI	0.17 ± 0.82	0.23 ± 1.18	0.640
zWeight	0.04 ± 0.82	0.12 ± 1.2	0.645
zHeight	−0.16 ± 0.92	−0.12 ± 1.03	0.761
zBMI > 0 (median), %	51.10	48.40	0.661
Overweight, %	5.8	7.4	0.867
Obesity, %	6.5	6.6	0.867
Overweight and obesity, %	12.2	14	0.683

Data are expressed as mean ± standard deviation or as percentage. zBMI: body mass index standard deviation score, zWeight: weight standard deviation score, zHeight: height index standard deviation score, zBMI > 0 (median) individuals with a zBMI value greater than 0, which is the median value of the total sample. * Significant difference between groups.

**Table 2 ijerph-18-06569-t002:** Evolution of the body mass index z-score (zBMI) in the participants of the control and intervention groups during the study.

zBMI	Control					Intervention				
	n	Baseline	Final	Change	*p*	n	Baseline	Final	Change	*p*
1st year	138	0.17 ± 0.82	0.14 ± 0.96	−0.03	0.261	121	0.23 ± 1.18	0.10 ± 0.99	−0.13	0.002 *
2nd year	132	0.16 ± 0.83	0.17 ± 1.03	0.01	0.423	115	0.24 ± 1.21	0.14 ±1.05	−0.10	0.021 *

Data are expressed as mean ± standard deviation. The baseline value at the beginning of the study is compared with the final value at the end of the first year in the first row and the final value at the end of the second year in the second row. * Significant change compared to baseline (before the intervention).

**Table 3 ijerph-18-06569-t003:** Evolution of body mass index z-score (zBMI) in the participants of the control and intervention groups stratified according to the baseline zBMI of the students.

Subgroup	zBMI	Control					Intervention				
		n	Baseline	Final	Change	*p*	n	Baseline	Final	Change	*p*
zBMI < 0	1st year	68	−0.47 ± 0.33	−0.52 ± 0.42	−0.05	0.296	63	−0.48 ± 0.31	−0.48 ± 0.45	0.00	0.401
	2nd year	67	−0.47 ± 0.33	−0.50 ± 0.41	−0.03	0.511	59	−0.49 ± 0.32	−0.43 ± 0.52	0.06	0.764
zBMI > 0	1st year	70	0.80 ± 0.65	0.78 ± 0.89	−0.02	0.64	58	1.00 ± 1.30	0.72 ± 1.05	−0.28	0.001 *
	2nd year	65	0.83 ± 0.66	0.86 ± 1.03	0.03	0.694	56	1.01 ± 1.32	0.73 ± 1.15	−0.28	0.002 *

Data are expressed as mean ± standard deviation. The baseline value at the beginning of the study is compared with the final value at the end of the first year in the first row and the final value at the end of the second year in the second row. * Significant change compared to baseline (before intervention).

**Table 4 ijerph-18-06569-t004:** Evolution of the prevalence of overweight and obesity in the participants of the control and intervention groups during the study.

		Control				Intervention			
		Baseline	Final	Change	*p*	Baseline	Final	Change	*p*
Overweight and obesity %	1st year	12.2	15.8	3.6	0.227	13.9	13.9	0	1
	2nd year	12.2	20.1	7.9	0.027 *	13.9	18	4.1	0.302

Data are expressed as percentage. The baseline value at the beginning of the study is compared with the final value at the end of the first year in the first row and the final value at the end of the second year in the second row. * Significant change compared to baseline (before intervention).

## Data Availability

The data presented in this study are available on request from the corresponding author. The data are not publicly available due to privacy and ethical restrictions.

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
