# Peer review of "Early Nutritional Education in the Prevention of Childhood Obesity"

_ijerph, 2021, doi:10.3390/ijerph18126569_

Round 1
Reviewer 1 Report
Thank you for the opportunity to review Early Nutrition Education in the Prevention of Childhood Obesity. This manuscript reports on a randomized controlled trial of a nutrition education intervention to prevent pediatric obesity in preschool aged children. The study involves an important area of intervention and provides interesting information concerning the potential impact of a low-intensity intervention. However, there are significant concerns that severely hinder the potential impact of the findings.
Introduction
- Use past tense when referring to previous research (i.e., page 1 lines 34-37).
- Consider rewording the last sentence of the first paragraph (lines 37-39) to clarify why is meant by the percentages presented.
- References should be updated and attention paid to ensuring that statements are appropriately cited (e.g., eight risk factors for obesity cite one study over 15 years ago and is not related to each of the risk factors cited including male sex).
- The authors provide sufficient justification for the target age range, but do not provide appropriate justification for the intervention being measured and rationale for school-based intervention.
- Please include specific hypotheses.
Methods/Results
- Are children present for intervention sessions?
- Variables and analytic section need more detail. Specifically, the “Statistical Analysis” section mentions qualitative variables (page 3, line 115), however no qualitative variables have been mentioned.
- Please clarify whether analyses accounted for clustering.
- How did the authors account for missing data? Consider intent to treat analyses.
- It is unclear why the authors did a median split of zBMI and how this may provide useful information, especially as dichotomizing a continuous variable decreases power and provides an artificial cut point.
- Overall, more specificity is needed in the statistical analysis plan.
Discussion
- Rationale for use of zBMI should be included in measures, rather than in discussion (page 6, lines 178-181).
- What is the benefit of a school-based intervention delivery when the intervention is specific to parents? Are there certain parents who may not be able to participate due to this constraint (i.e., are working during session)?
- It would be helpful to have a better sense of the parental involvement, average number of sessions attended, any demographic characteristics impacting ability to engage in the intervention or impacting zBMI change in children.
- At multiple points in the discussion, the authors discuss family-based treatment. If the authors consider this a family-based intervention for pediatric obesity, that should be clarified throughout the manuscript.
- Is the information on future interventions (page 7, lines 247-252) based upon information collected in the present study? It would be helpful for the authors to provide data on these intervention components to clarify how they might be better addressed in future interventions.
Reviewer 2 Report
Dear authors, congratulations on your great work. My suggestions and questions are in the manuscript.

Reviewer 3 Report
What were the inclusion criteria for the studies?
What were all the exclusion criteria from the study?
Were the children weighed on fasting status? at what time of day?
Has the bioethics committee approval been obtained
Were the children asked about their consent to participate in the study?
If I understand correctly, each parent and child participated in the various stages of the intervention. How often were the meetings held?
What does this research contribute to science?
Why this age range of participants exactly? What the authors suggested when selecting the group.
Round 2
Reviewer 1 Report
Thank you for the opportunity to review the revised manuscript, Early Nutrition Education in the Prevention of Childhood Obesity. The authors have made a number of changes that strengthen the overall manuscript. Some concerns remain, which are outlined below.
- Concerns remain about stating the intervention is “family-based,” especially considering the authors have clarified that the child was not present for the intervention. Family-based treatments for pediatric obesity involve the entire family, including both the parent and the child (e.g., Fitzgibbon et al., 2013 – “Family-based hip hop to health: Outcome results”). The treatment presented is a parent-only intervention.
Methods/Results
- It should be clarified in the text that children were not present for intervention sessions.
- There is still no description of the qualitative variables in the text. It appears that the authors may be misusing the term “qualitative” as there are no qualitative variables present in either the text or tables. If the authors are referring to percentage of children with overweight/obesity, this does not constitute a qualitative variable.
- Information regarding clustering and missing data in the response letter should be included in the analytic plan.
Reviewer 3 Report
The authors answered all questions and doubts of the reviewer.
I recommend publishing this manuscript
Author Response
Thank you very much for your appreciation, and for the recommendations that helped us optimize this work.